# Antibacterial Activity of Some Essential Oils/Herbal Extracts Against Bacteria Isolated from Ball Pythons (*Python regius*) with Respiratory Infections

**DOI:** 10.3390/antibiotics14060549

**Published:** 2025-05-28

**Authors:** Corina Pascu, Viorel Herman, Luminita Costinar, Corina Badea, Valentin Gros, Georgeta Stefan

**Affiliations:** 1Department of Infectious Diseases and Preventive Medicine, Faculty of Veterinary Medicine, University of Life Sciences “King Mihai I”, 300645 Timisoara, Romania; viorel.herman@fmvt.ro (V.H.); corina.badea@usvt.ro (C.B.); 2Department of Microbiology, Faculty of Veterinary Medicine, University of Life Sciences “King Mihai I”, 300645 Timisoara, Romania; valentingros@usvt.ro; 3Clinical Sciences 1 Department, Faculty of Veterinary Medicine, University of Agronomic Sciences and Veterinary Medicine Bucharest, 105 Splaiul Independentei, 5th District, 050097 Bucharest, Romania; georgeta.stefan@fmvb.usamv.ro

**Keywords:** ball pythons, respiratory disease, antimicrobial resistance, essential oils/herbal extracts

## Abstract

Background: Respiratory diseases are among the main causes of morbidity and mortality in captive reptiles. In Romania, pneumonia is a frequently observed illness affecting pet reptiles. Key factors contributing to the high incidence of pneumonia include inadequate animal husbandry, poor nutrition, and insufficient hygiene practices. Bacteria may act as primary pathogens or as facilitators of disease severity. Methods: This study investigates bacterial strains from multiple genera and species (*Chryseobacterium* (*C.*) *indologenes*, *Staphylococcus* (*S.*) *epidermidis*, *Escherichia* (*E.*) *coli*, and *Pseudomonas* (*P.*) *aeruginoasa*) from six ball pythons regarding their antibiotic susceptibility and the effect of essential oils. Bacteria were isolated from the lower respiratory tract, displaying clinical signs of pneumonia. All isolates were tested with essential oils (lemongrass, oregano, rosemary, and sage) and a grapefruit seed extract (GSE) at different dilutions. Results: The incidence of *Chryseobacterium indologenes* was highest (3 isolates/12 samples, 25%), followed by *E. coli* and *Staphylococcus epidermidis* (2/12 each, 16.6%), and *Pseudomonas aeruginoasa* (1/12, 8.3%). Resistance profiling to different antibiotic classes revealed that all isolates (eight) were resistant to multiple antibiotics tested by us. All isolates were resistant to β-lactams and fluoroquinolones. One strain of *E. coli* exhibited intermediate resistance to quinolone and penicillin. All strains were categorized as multidrug-resistant. GSE showed antibacterial activity against all isolates. Conclusions: Wanting to deepen our understanding of the microorganisms that can infect ball pythons and recognizing that all isolated bacteria have zoonotic potential, this paper highlights some common issues faced by exotic animal owners and suggests that treatments should also include the use of essential oils.

## 1. Introduction

Numerous infectious and parasitic pathogens have been isolated from reptiles with respiratory diseases. Although the initial approach to treating snake patients follows the same principles as with domestic animals, advanced diagnostic methods, test interpretation, and therapeutic protocols must be adapted due to the unique morphology of snakes’ respiratory systems [1,2]. In Romania, studies on herpetofauna health are in the early stages, with very little information available on the infectious pathology of reptiles, such as ball pythons, while many studies focus primarily on turtle pathology [3]. Furthermore, in Romania, the surveillance of herpetofauna health is poorly structured, lacking coherent national programs for the epidemiological monitoring of pathogens in reptiles. Despite the inclusion of several species under European conservation directives [4], veterinary health aspects remain marginalized, with no standardized diagnostic protocols or centralized data collection systems [5,6]. This deficiency is further exacerbated by institutional fragmentation among environmental, public health, and veterinary authorities, limiting the implementation of an integrated “One Health” approach to mitigate zoonotic risks associated with herpetofauna.

Respiratory infections in reptiles are common, and their incidence may be influenced by respiratory or systemic parasitism, unfavorable environmental temperatures or humidity, insufficient ventilation, unsanitary conditions, co-morbidities, and malnutrition. Open-mouth breathing, nasal or glottal discharge, and dyspnea are common signs. *E. coli* and *Pseudomonas* species are commonly isolated, but other microorganisms may also be involved [7,8,9,10].

In reptiles, *Pseudomonas aeruginosa* is considered both a normal microflora of the oral cavity and intestinal tract as well as an opportunistic pathogen, being responsible for various clinical symptoms. Under certain conditions—particularly when strains are highly pathogenic—*P. aeuruginosa* has been linked to significant morbidity and mortality in infected animals. This is largely due to its ability to develop antibiotic resistance and to express multiple virulence factors, *Pseudomonas* being isolated from snakes with ulcerative stomatitis, pneumonia, cutaneous lesions, and septicemia [11,12]. Reptiles are considered sources of contamination for humans because the owners may come into contact with feces or through the production of wounds and scratches on humans by the infected animal [13,14].

*Chryseobacterium indologenes* is a Gram-negative bacterium and is found sporadically in the environment, having often been detected in water, soil, food sources, waste, and aquatic environments. *Chryseobacterium* species have been reported to cause infections in neonatal infants, pregnant women, or immunocompromised patients [15,16,17]. The isolation of *Cryseobacterium idologenes* from reptiles (different python species) that are clinically healthy or with different symptoms is mentioned in few studies in the recent literature [18,19].

Data on the *E. coli* isolated from reptiles (reptilian *E. coli*—RepEC) are very limited [20,21]. It is known from the literature that *E. coli* can be a commensal bacteria, present in the digestive tract of mammals and birds, but at the same time, there are also intestinal (IPEC) and extraintestinal (ExPEC) pathogenic strains. Although the reservoir of these strains is large and small ruminants, they have also been isolated from reptiles [13,14]. In a study conducted by Dec et al. [22] on *E. coli* isolated from pet reptiles, the highest proportion of *E. coli* strains was isolated from pet snakes (51.6%). Although the majority of reptilian *E. coli* strains belong to phylogroup B1 or C, which include non-pathogenic commensal strains, pathogenic strains of RepEC have also been identified, falling into phylogroup B2 [21].

Bacteria such as *E. coli*, *Pseudomonas* or *Cryseobacterium* are generally considered to be part of the commensal flora of reptiles. However, under certain conditions, they can express virulence factors, producing different symptomatologies. In addition, the zoonotic character of these bacteria is known, and they can cause infections in people coming in contact with reptiles, especially immunocompromised people.

Identifying the bacteria associated with respiratory disorders in reptiles is crucial for expanding our understanding of these microorganisms, as well as for understanding the etiology of secondary infections that can arise from animal management practices.

The treatment of infections in reptiles is difficult, both in terms of the route of administration and the choice of antimicrobial substance, as the potential toxic effects of antimicrobials on reptiles are well known. In addition, there is limited knowledge about the pharmacokinetics of antimicrobials in snakes. Although some antibiotics are recommended, the phenomenon of resistance or multiple resistance has also been reported in reptiles. In these conditions, finding an alternative is salutary. Essential oils, used in alternative human medicine, have also begun to be used in veterinary medicine. Their use in snakes is little studied, as some essential oils or plant extracts are toxic to reptiles [7,8,9,10]. In this context, this study aims to provide new data on the effect of essential oils and GSE on bacterial strains isolated from reptiles.

## 2. Results

### 2.1. Identification of Bacterial Strains

The identification of all isolates grown on culture media were confirmed by API 20E, API 20NE, API Staph (bioMerieux, Marcy l`Etoile, France), and MALDI TOF-Ms. Glucuronidase and β-galactosidase production was also confirmed in *E. coli* isolates by a phenotypic test using Brilliance ESBL agar (Oxoid, Hampshire, UK). The presence of *Chryseobacterium indologenes* was recorded in 3 of 12 samples (25%), *E. coli* and *Staphylococcus epidermidis* (2/12 each, 16.6%), and *Pseudomonas aeruginoasa* (1/12, 8.3%) (Table 1). This study, although innovative, has an important limitation: it is a pilot study and has a limited number of samples and isolates, respectively, thus restricting the generalizability of findings.

### 2.2. Antimicrobial Susceptibility Testing

In both human and veterinary medicine, the percentage of multidrug-resistant (MDR) bacteria is constantly increasing due to the abuse and irrational use of antibiotics, and the number of nosocomial infections caused by these bacteria is also increasing.

Bacteria isolated from ball pythons with respiratory symptoms exhibited varying degrees of resistance against most antimicrobials, including cephalosporins (cefaclor and ceftiofur), beta-lactams (ampicillin and penicillin), and fluoroquinolones (ciprofloxacin and enrofloxacin) (Table 2).

Table 3 indicates that all bacterial strains isolated from pet reptiles (ball pythons) exhibited multidrug-resistance (MDR), defined as resistance to three or more antimicrobial agents. Isolated bacteria showed sensitivity to phenicols (florfenicol), sulfamethoxazole, tetracyclines (doxycycline), and aminoglycosides (amikacin).

*C. indologenes* isolates are characterized by remarkable antibiotic resistance, being resistant to 6 out of 10 antibiotics tested, respectively, to ampicillin, cephalosporins, penicillin and fluoroquinolones. *E. coli* isolates have different responses to antibiotics, one of them presenting resistance only to three antibiotics (ampicillin and cephalosporins). *S. epidermidis* have the same resistance profiles, being resistant at five of 10 antibiotics tested. *P. aeruginosa* presents the highest resistance, being resistance to seven antibiotics.

It can be concluded that the resistance to ampicillin and cephalosporins (cefaclor and ceftiofur) was registered in all (100%) isolates, at fluoroquinolones (enrofloxacin and ciprofloxacin) the resistance was found in 87.5%, penicillin 62.5%, and amikacin 12.4%.

### 2.3. Minimum Inhibitory Concentration of Essential Oils

Table 4 presents the effect of different dilutions of essential oils/vegetal extract on bacteria isolated from the oral cavity of ball pythons.

For these determinations, a disk impregnated with enrofloxacin (5 µg) (Oxoid) was used as a positive control, and sterile BHI (Oxoid) culture medium was used as a negative control.

The interpretation of the antibacterial activity was based on the diameter of the zone of inhibition, according to the indicative classifications in the literature [24,25]. Inhibition zones >20 mm were considered as an indication of high activity, between 15 and 20 mm—medium activity, and <15 mm—low/inactive activity.

In this study, from six plant extracts known for having antibacterial activity, only GSE showed an antibacterial effect. This effect was most evident in *C. indologenes*. The GSE’s antibacterial activity decreases with increasing dilution, indicating a clear dose-effect relationship. GSE exhibits a broad spectrum of antibacterial activity, inhibiting both Gram-negative (*E. coli*, *P. aeruginosa*, *C. indologenes*) and Gram-positive (*S. epidermidis*) bacteria.

*P. aeruginosa* is the most resistant of the strains tested—inhibition is absent (NI = no inhibition) at dilutions greater than 1/32. *C. indologenes* is the most sensitive strain—the zone of inhibition remains detectable even at high dilutions (7 mm at 1/256).

Oregano essential oil has a moderate activity at 1/2 dilution, and this was the only dilution that had an activity on *C. indologenes*. LGEO, REO, and SEO have no inhibition activity of bacterial growth at any of the dilutions we tested, which proves that *C. indologenes* is resistant to these essential oils.

Only GSE and oregano essential oil had an inhibitory effect on *E. coli* isolates. The essential oils of lemongrass, rosemary, and sage do not appear to have an inhibitory action on this bacterium.

Oregano essential oil produced an inhibition at 1/2 dilution, but the diameter classified the result as a moderate effect. At the other concentrations (1/4–1/256) it does not influence the growth and multiplication of *E. coli*. Based on these results and the lack of other information in the literature on the bacterial flora isolated from reptiles, we can state that this oregano essential oil has a questionable influence on *E. coli*.

The lemongrass, rosemary, and sage essential oils are inadequate as a possible alternative treatment for *E. coli* infection.

The diameter zones for grapefruit seed extract observed for *S. epidermidis* allow results to be framed as moderate activity only. The other studied essential oils did not influence the growth of the *S. epidermidis* bacterium. Although it was noted that oregano oil acts on the Gram-negative bacteria *Escherichia coli* and *C. indologenes*, on *Staphylococcus*, which is a Gram-positive bacterium, it was observed that it had no inhibitory effect. The culture showed typical morphological and growth characteristics; it grew and developed normally. 

For *P. aeruginosa*, colonies developed normally on all Petri dishes and pigment production was not inhibited. The only dish where interaction between the tested essential oil and the *P. aeruginosa* was observed corresponds to the GSE, but the *Pseudomonas* strain is resistant to all dilutions of the extract. However, it can be observed that at dilutions of 1/2 and 1/4, diameter zones can be classified as moderate, and the synthesis of the pigment characteristic for this bacterium is slightly inhibited.

It is known from the literature that the pigment produced by *Pseudomonas* (pyocyanin) plays an important role in the pathogenesis of *Pseudomonas* infections. Several effects of pyocyanin have been described: cytotoxicity, neutrophil apoptosis, ciliary dysmotility, and pro-inflammatory effects. This has been demonstrated by the fact that mutant strains of *Pseudomonas* which have lost the ability to perform pigmentogenesis are much more attenuated. The remarkable toxic properties of the pigment can be demonstrated by the large number of target organisms of *Pseudomonas*, including other bacteria, fungi, insects, nematodes, and plants [26].

For the other essential oils, it was found that none of the studied oils affected pigment synthesis.

### 2.4. Essential Oils Analysis

The quantitative compounds of the essential oils (lemongrass, rosemary, sage, and oregano) and grapefruit seed extract measured through GS-MS and HPLC are presented in Table 5 and Table 6, and in Table 7, respectively.

The chemical composition of the analyzed essential oils reveals the presence of several well-documented bioactive compounds, whose relative concentrations can explain the differences observed in antibacterial efficacy.

Oregano essential oil demonstrated the highest antibacterial potential, which can be attributed to its high content of carvacrol (64.70%) and thymol (3.60%). These phenolic monoterpenes are known for their strong antimicrobial effects, primarily through mechanisms involving disruption of bacterial membranes, ion leakage, and interference with metabolic processes. In addition, p-cymene (14.94%), although not strongly active by itself, may enhance the activity of carvacrol and thymol by increasing membrane permeability, supporting a synergistic effect [27,28].

Lemongrass essential oil showed high concentrations of citral components (mixture of *β-citral*—28.26% and *α-citral*—43.43%), which are potent antimicrobial aldehydes. Citral exerts its effects by altering membrane fluidity and inhibiting key enzymatic pathways [28].

In contrast, rosemary essential oil was dominated by limonene (49.31%), camphor (14.61%), and α-pinene (11.95%). While these monoterpenes exhibit some antimicrobial activity, their potency is generally lower compared to phenolic or aldehydic constituents [27,28,29].

Sage essential oil was characterized by a high percentage of linalyl anthranilate (77.34%) and linalool (17.06%). Although linalool has moderate antimicrobial properties, the dominance of linalyl anthranilate, an ester with limited antimicrobial efficacy, may explain the relatively weak antibacterial activity observed for this oil [29].

The chemical analysis of GSE revealed a complex profile composed predominantly of aliphatic amines, long-chain hydrocarbons, and oxygenated monoterpenes. The most abundant constituent was 3-(N,N-dimethylmyristylammonio)propanesulfonate (27.99%), a quaternary ammonium compound with known membrane-disruptive activity. This compound likely plays a central role in the antibacterial efficacy observed across all tested bacterial strains, particularly at lower dilution ratios. Other major constituents included 1-dodecanamine, N,N-dimethyl- (15.67%), 1-hexadecene (12.24%), benzenemethanamine, N,N-dimethyl- (10.06%), and 1-tridecanamine, N,N-dimethyl- (9.08%). These tertiary aliphatic amines and long-chain hydrocarbons exhibit amphiphilic characteristics that are often associated with surfactant-like properties. Their presence may contribute to the destabilization of bacterial membranes by enhancing permeability or interfering with membrane-associated proteins. Although present at a moderate concentration (6.83%), limonene—a monoterpene hydrocarbon known for its antimicrobial effects—may also contribute to the observed inhibition. Limonene is capable of disrupting cell membranes and inhibiting energy metabolism, and its presence in the GSE matrix may act synergistically with amines and quaternary compounds [30,31,32].

High-performance liquid chromatography (HPLC) analysis of the polar fraction of GSE revealed the presence of several bioactive phenolic compounds and antioxidants, including ascorbic acid, flavonoids, and phenolic acids (Table 7).

The most abundant compound was p-coumaric acid (*cis*-form, retention time = 12.66 min), detected at a concentration of 6.87 µg/mL. This hydroxycinnamic acid is known for its antimicrobial, antioxidant, and anti-inflammatory properties and may synergize with other phenolic acids in destabilizing microbial membranes and inhibiting enzymatic activity [31,32].

Naringin, a characteristic flavonoid of citrus seeds, was present at 3.82 µg/mL, reflecting the grapefruit origin of the extract. Naringin is known for its bacteriostatic effects, particularly against Gram-negative strains, via interference with quorum sensing and efflux pump inhibition.

Ascorbic acid was quantified at 2.31 µg/mL, consistent with the known antioxidant profile of citrus-derived matrices. While not directly antimicrobial, ascorbic acid contributes to oxidative stress regulation, potentially enhancing the stability and bioactivity of other polyphenols.

Quercetin, a flavonol with broad antimicrobial and anti-inflammatory activity, was identified at 1.15 µg/mL, followed by caffeic acid (0.61 µg/mL) and ferulic acid (0.50 µg/mL), both of which are known to exert antimicrobial effects through membrane perturbation and oxidative damage mechanisms [30,31,32].

## 3. Discussion

This study represents the first report from Romania of a *C. indologenes* infection in the royal python (*Python regius*), a reptile species commonly kept as a pet. The pathology of pet reptiles, in particular snakes, is attracting increasing interest due to the close relationship between keepers and animals and the zoonotic risk posed by commensal or opportunistic bacteria carried by reptiles. *C. indologenes* is a bacillary, aerobic, non-motile, Gram-negative, non-fermenting bacterium belonging to the *Flavobacteriaceae* family. Initially regarded as an opportunistic pathogen in nosocomial human infections, recent evidence indicated its occurrence in a broader range of ecological niches, including associations with exotic animals such as reptiles. This ecological expansion reflects its high adaptability and highlights the underexplored role of *C. indologenes* in veterinary and herpetological medicine [18,29,30].

The primary reservoirs of *C. indologenes* include moist environments and biofilms, often found on medical equipment, water systems, and soils [15,18,30]. These habitats are also relevant in reptile husbandry, where humid terrarium conditions, standing water, and organic debris can support long-term bacterial persistence. Biofilm formation on enclosure surfaces, misting systems, or water bowls may contribute significantly to colonization and transmission.

Although *C. indologenes* is traditionally viewed as a non-specific environmental pathogen [15,16,17], recent isolations from snakes, turtles, and lizards suggest a growing ability to colonize reptilian skin, mucous membranes, and possibly respiratory epithelium. Mesophilic growth preference (20–37 °C) makes terrarium environments particularly favorable. Reptiles may act as intermediate hosts or asymptomatic carriers, especially under conditions of stress or immunosuppression.

There were several identified factors (intrinsic and extrinsic) that may contribute to the emergence of *C. indologenesi* as a reptile-associated pathogen: chronic stress (due to transport, handling, poor environmental conditions), prior broad-spectrum antibiotic use, persistent biofilms in aquatic systems, enclosure surfaces, minor lesions or traumas, and parasitic or viral infections (leading to immune suppression) [33,34,35,36]. All these factors, along with resistance to multiple antibiotics, suggest that *C. indologenes* may represent an emerging threat in captive reptiles [37].

Our research is similar to the research conducted by Yeon-Sook Jho et al. [38] in 2011. In this study, it is reported that *Pseudomonas* spp., *E. coli*, *C. indologenes*, and many other bacterial genera can be isolated and identified in the oral cavity and cloaca in snakes, more precisely in the *Burmese python*. Compared to Schmidt et al. [4], who predominantly isolated *Salmonella* spp. bacteria from snakes with respiratory diseases, from the *Pythonidae* family, we did not isolate any *Salmonella* from ball pythons tested. Although at first glance there seems to be a discrepancy between the bacteria isolated in this study and those isolated in other studies, there is an explanation for this, namely, the type of samples and where they were collected. The samples collected in this study were oral swabs from pythons with respiratory symptoms. *Salmonella* is known to be a common colonizer of the GI tract of reptiles, and its isolation is performed using cloacal swabs, fecal samples, or tissue biopsies.

Regarding the antimicrobial sensitivity of the isolated strains, a different behavior was observed, with multiple resistances being recorded in most of the strains isolated from the oral cavity of ball pythons.

A study by Foti et al. [39] on 65 samples collected from the oral cavity of 15 different species of reptiles kept in captivity showed the resistance of *Pseudomonas* bacteria to some antibiotics. The isolation rate of *Pseudomonas aeruginosa* from cultures was variable between the different studied species, but all isolates showed frequent antibiotic resistance. The highest degree of resistance was to imipenem cilastatin (100%), penicillin G (95.6%), sulfamethoxazole (93.5%), erythromycin and vancomycin (91.3%), doxycycline (85.3%), and tetracycline (80.4%). Resistance was also frequently observed for cephalexin, cephalothin, rifampicin (78.3%), ampicillin (71.7%), and nalidixic acid (69.6%). Our results are similar to the results obtained in the previously mentioned study.

Another study suggested a higher prevalence of *Pseudomonas aeruginosa* in captive snakes than in wild specimens, but there are insufficient data to make a correlation between captivity and the isolation frequency of *Pseudomonas aeruginosa* in snakes [8].

Other authors have reported methicillin-resistant *Staphylococcus aureus* (MRSA) as the main cause of abscess in a snake with symptoms of respiratory failure [33,34].

In 2021, in Turkey, Tamai et al. [18] reported a case study with oral abscesses caused by this bacterium in a three-year-old ball python. *C. indologenes* isolated from python oral abscesses taken in the Tamai study were sensitive to ceftazidime, minocycline, and tetracycline. Our results are similar to those obtained by Tamai regarding the sensitivity of *Chryseobacterium* strains to the tetracycline group (doxycycline), but a remarkable resistance to cefaclor and ceftiofur was observed in our study.

From the literature, it is known that there is a close genetic relationship between the human-isolated strains and animal-isolated strains [35,39,40,41,42].

The antimicrobial resistance patterns of *C. indologenes* strains isolated by us in this study are different from those of strains isolated from human infections reported by other researchers [3,7,17].

This multidrug-resistance observed in this study may be the result of the action of various factors. One of them is anthropogenic antibiotic use. Snakes kept in captivity may acquire resistant bacterial strains through direct or indirect exposure to antibiotics used in other animals or humans in the same environment [43,44]. Another factor is represented by environmental contamination. Aquatic and terrestrial ecosystems contaminated with wastewater, improperly disposed antimicrobials, and livestock runoff act as reservoirs of resistant bacteria and resistance genes.

Considering these aspects, we recommend before administering treatment to reptiles (snakes) carrying out bacteriological examinations and carrying out the antibiogram to determine exactly the antibiotic of choice, knowing that these bacterial strains multiresistant to antibiotics can reach humans either by direct contact with pets or by eating these reptiles, especially in geographic areas where reptiles are commonly considered as a food supply.

Since ancient times, natural products have been used to treat various infections in both humans and animals. Natural-based products underlie the development of almost all the medicinal products on the market today. Some authors suggest that natural products are better accepted by the body than chemical products. The treatment of companion animals, particularly reptiles, poses significant challenges due to the unique anatomy and physiology of these animals, as well as the specialized method of administering medication [27].

Unfortunately, in recent years, there has been a marked increase in the percentage of antibiotic-resistant bacteria in pet reptiles. Interest in natural products as replacements for traditional antibiotics to combat multidrug-resistant pathogens has greatly increased. The antibacterial effect of bee honey and the antibacterial effects of plant extracts such as black pepper extract, grapefruit seed extract (GSE), and other essential oils have been proven in numerous papers [23,27,30,40].

We therefore focused on natural products, in particular plant extracts (GSE), but also some essential oils. GSE is a biological material derived from citrus fruits containing various bioactive components such as polyphenols, flavonoids, ascorbic acid, organic acid, etc., which are believed to be responsible for the potent antimicrobial and antioxidant activity [27,34,39,45].

GSE is a natural product that had the best in vitro effect on *C. indologenes* and *E. coli.* For the other isolates—*S. epidermidis* and *P. aeruginoasa*—the effect was moderate only. We consider that further studies are needed to understand and study this natural product in vivo.

Grapefruit seed extract is obtained by first turning the grapefruit seeds and pulp into a highly acidic liquid. This liquid has many polyphenolic compounds, including quercitin, helperidine, kaempferol glycoside, neohelperidine, naringin, apigenin, rutinoside, and poncirin, which are believed to contribute to its antimicrobial and antioxidant effects. Polyphenols are unstable and are transformed into more stable substances, namely quaternary ammonium compounds [30,31,32,46]. These grapefruit-derived quaternary ammonium compounds have broad-spectrum antimicrobial activity, as can also be seen from this study we performed on bacteria isolated from ball pythons [28,29,47,48]. According to the literature, the effectiveness of GSE is influenced mainly by its polyphenol concentration, particularly the presence of citrus flavonoids like naringin, limonin, and others [30].

It is known from the literature that many commercial GSE products contain synthetic compounds added for preservation that have a well-known antimicrobial effect in addition to the natural, GSE-specific compounds [30,31,32]. In this context, the question may be raised as to what extent the antimicrobial effect of commercial GSE products is due to the natural compounds and not to the added synthetic compounds. An answer to this question can be considered using the study by Murgia et al. [49] in which it was demonstrated that the antimicrobial effect of GSE is due to natural compounds, having used a commercial GSE chemical-free product in the study.

Comparing the efficacy of essential oils and GSE, it can be observed that in in vitro conditions, GSE demonstrated the strongest antibacterial effect on *C. indologenes* and *E. coli*. In contrast, the efficacy on *P. aeruginosa* and *S. epidermidis* was moderate. The essential oils tested (oregano, lemongrass, rosemary, sage) had no significant inhibitory effect on the bacterial strains studied, which does not support their therapeutic use in pythons.

The efficacy of GSE is attributed to its high content of polyphenols, flavonoids, and quaternary ammonium compounds, which possess broad-spectrum antimicrobial activity.

To the best of our knowledge, this is the first study from our country to highlight the antibacterial action of essential oils in the alternative treatment of diseases encountered in pet reptiles (*Python regius*).

Our research did not focus on the biochemical analysis of essential oils or GSE to identify the specific compounds with antimicrobial activity, as this area falls outside our primary field of expertise and is already well-documented in the literature. The objective of testing these products was to evaluate their antimicrobial efficacy and to explore their potential use as alternative treatments for infections caused by multidrug-resistant bacteria.

The bacterial strains used in this study were isolated from ball pythons (*Python regius*), a species that requires a distinct therapeutic approach compared to other animal species due to their high sensitivity to certain antimicrobial agents and the frequent occurrence of adverse effects. Consequently, this study offers a potential therapeutic alternative for veterinary practitioners dealing with such challenging infections.

The pronounced sensitivity of *Python regius* to many synthetic antimicrobials necessitates the exploration of less toxic yet effective treatment options. The antimicrobial efficacy observed in this study aligns with previously published data on essential oils’ effects on human and animal pathogens.

The results of this study indicate the need for routine susceptibility testing prior to treatment in pet reptiles. The empirical administration of antibiotics in the absence of such tests may favor the development and spread of multidrug-resistant bacteria with major zoonotic impact.

Given the increased sensitivity of reptiles to certain synthetic antibiotics and the incidence of adverse reactions, alternative therapies based on natural products could be a viable solution. GSE has shown promising potential as an antimicrobial agent in this context, but further studies are needed to address pharmacokinetics, safe profile, and in vivo efficacy, as well as appropriate formulation for administration to reptiles.

This study, although innovative, has some limitations. It is a pilot study and was performed on a limited number of specimens of pythons and isolates. It does not perform a detailed biochemical analysis of GSE and essential oils, and testing was limited to in vitro conditions with no pharmacological and toxicological evaluations.

It was impossible to clearly determine the possible contribution of the synthetic compound in the commercial products tested. Extending the research to a larger sample and including other reptile species could provide more conclusive results and contribute to validating the use of alternative treatments in exotic veterinary medicine.

## 4. Materials and Methods

Sampling was performed in March 2023 at the Faculty of Veterinary Medicine Timisoara, Romania. Samples were taken from 6 ball pythons (*Phyton regius*), two males and four females, aged between 8 months and 6 years old, which were kept in a terrarium by a reptile breeder in Timisoara, respecting the welfare of the animals and using the isolation technique suitable for this reptile species. The symptoms observed were anorexia, lethargy, difficulty breathing (wheezing), abundant secretions in the oral cavity, and weight loss. The diagnosis of respiratory infection was confirmed by radiologic examination, with pneumonia being observed in 4 of the pythons, and accumulation of fluid in lungs and air sacs in all examined pythons. No antimicrobial treatment was administered to the pythons before presentation at the clinic.

To ensure diagnostic accuracy, all samples were collected aseptically using sterile swabs from the python’s oral cavities. The samples were immediately transferred into sterile tubes with transport media AMIES (VWR, Carnaxide, Portugal). Samples were labeled appropriately and processed within 2 h post collection.

### 4.1. Bacterial Isolation

Oral swabs were spread on different culture media such as Baird Parker Agar (Oxoid, Hampshire, UK), Columbia blood agar (Oxoid, Hampshire, UK), and Manitol Salt Agar (Oxoid, Hampshire, UK), according to the manufacturers’ instructions. Plates were incubated at 37 °C for 18–24 h under aerobic conditions, with longer incubation up to 72 h for *Cryseobacterium.*

Samples were inoculated on culture media with 5% sheep blood agar (Oxoid, Hampshire, UK) and incubated at 37 °C for 18–24 h under aerobic conditions. Standard procedures were used to identify bacterial isolates, including the Gram staining method and colony morphology. Primary inoculation was performed according to the procedure known in the literature [23].

Phenotypic characterization and taxonomic classification were performed using the API 20NE, API 20E, and API Staph (BioMérieux, Marcy l’Étoile, France). To confirm these results, the isolated colonies were reanalyzed using MALDI TOF-Ms. Proteomic identification with the MALDI-ToF-Ms technique was realized based on the related methodology described below:Each dispersion from the initial plate was examined by at least 2 examiners, and one sample from each isolated colony with individual morphological aspect, resulting after incubation at 37 °C for 18–24 h, was prelevated and applied on the MALDI-TOF-Ms plate in duplicate;After drying, it was applied to 1 µL matrix 4-HCCA (acid α-Cyano-4-hydroxycinnamic, α-Cyano-4-hydroxycinnamic acid) and we waited until it was dry again. The plate was supposed for identification using the MALDI-TOF-Ms Bruker mark, model autoFlex Speed (Bremen, Germany).

Brilliance ESBL agar (Oxoid, Hampshire, UK) was used to detect specific biochemical characteristics of *E. coli* strains.

Interpretation of MALDI-TOF data was performed as follows: scores of ≥2.0 were accepted for species assignment and scores of ≥1.7 but <2.0 were accepted identified to the genus levels. Scores below 1.7 were considered unreliable. The following databases were used for isolate identification: MALDI Biotype Reference Library: MBT Compas Library Revision E, and MBT 7854 MSP Library (Bruker Daltonics, Bremen, Germany).

### 4.2. Antimicrobial Susceptibility Testing

Antimicrobial activity was performed using the disk-diffusion Kirby–Bauer method using purified cultures, on Mueller-Hinton agar (Oxoid, Hampshire, UK). The indications of CLSI [23] were used for the interpretation of the test after incubation at 37 °C for 24 h. This method is a flexible and relatively cheap technique, used in the laboratory [18,23,50].

The following antibiotics were used: Ampicillin (AMP; 10 μg) (Oxoid, Hampshire, UK), Amikacin (AMK 30 μg) (Oxoid, Hampshire, UK), Cefaclor (CCL 30 μg) (Oxoid, Hampshire, UK), Ceftiofure (EFT 30 μg) (Oxoid, Hampshire, UK), Ciprofloxacin (CIP 5 μg) (Oxoid, Hampshire, UK), Doxycycline (DO 30 μg) (Oxoid, Hampshire, UK), Enrofloxacin (ENR 5 μg) (Oxoid, Hampshire, UK), Florfenicol (FFC 30 μg) (Oxoid, Hampshire, UK), Penicillin (P 10 UI) (Oxoid, Hampshire, UK), and Sulfamethoxazole (SXT 5 μg) (Oxoid, Hampshire, UK).

The reference strains *E. coli* ATCC 12814 and *Staphylococcus intermedius* NCTC 13924 were used as quality control according to CLSI recommendations.

### 4.3. Determination of the Minimum Inhibitory Concentration (MIC) by Broth Microdilution Method

MIC can be obtained using two methods—broth dilution or agar dilution. These tests are reference methods, used to evaluate the accuracy of other test methods [33,43,50]. Additional equipment is required to prepare stock solutions of antimicrobial agents, which are then dispensed into test tubes (macro-dilution), well plates (micro-dilution), or solid media. Each compartment contains essential oils in a different standard concentration.

The MIC level can be determined based on the level of visible growth of the bacterial inoculum. To carry out the research presented in this work, the broth dilution method was used, using sterile well plates (micro-dilution).

Essential oils with antibacterial effects studied and known in the literature to be beneficial for animals were chosen. These essential oils are not known to be toxic or harmful to reptiles.

We used the following essential oils/herbal extracts:Citrosol (grapefruit seed extract—GSE), Interherb Targu Mures, Romania;Lemongrass essential oil (*Cymbopogon citratus* essential oil—LGEO), Solaris, Bucharest, Romania;Oregano essential oil (*Origanum vulgare* essential oil—OEO), Solaris, Bucharest, Romania;Rosemary essential oil (*Rosmarinus Officinalis* essential oil—REO), Solaris, Bucharest, Romania;Sage essential oil (*Salvia Officinalis* essential oil—SEO), Solaris, Bucharest, Romania.

An aliquot (100 µL) was taken from the pure bacterial culture and dissolved in 4 mL of sterile saline until a turbidity of 0.5 on the McFarland scale was obtained, which corresponds to a bacterial concentration of 1–2 × 10^8^ CFU/mL. From this suspension, 50 microliters was transferred into 10 mL of nutrient broth (Oxoid, Basingstoke, UK). The final inoculum concentration was 1–5 × 10^5^ CFU/mL. It is recommended to use the inoculum within 15 min after obtaining it.

Performing Microdilution in Plates

The arrangement of the test elements in the wells of the plate was performed as follows:Well A1—200 microliters of essential oil;Wells A2 to A9—100 microliters of BHI broth (Oxoid, Basingstoke, UK) in each well;The serial dilution was carried out—for this, 100 microliters of the pure extract was taken and transferred to serial wells, until the 8th dilution was reached, i.e., the 1/256 dilution. (1/2, 1/4, 1/16, 1/32, 1/64, 1/128, 1/256);This serial dilution was performed for the five analyzed essential oils.

Petri dishes with Muller-Hinton agar (Oxoid, Basingstoke, UK) were used. Each plate was inoculated with bacterial culture, then all plates were allowed to dry in the thermostat (approximately 3–5 min). After the surface had dried, wells with a diameter of 5 mm were cut in the agar and 20 microliters of each diluted essential oil was added into these wells. Petri dishes were incubated with the lid facing up at 37 °C for 24 h. A Petri dish was used for each essential oil/herbal extract.

Antimicrobial activity was assessed by measuring the diameter (mm) of the growth inhibition zone around the well into which the essential oil dilution was introduced for eight concentrations tested (from 1/2, 1/4, 1/8, 1/16, 1/32, 1/64, 1/128, 1/256) against strains isolated from the ball pythons.

### 4.4. Gas Chromatography–Mass Spectrometry (GC–MS)

The volatile oils were analyzed by gas chromatography using a Shimadzu QP 2010 Plus apparatus (Columbia, SC, USA). For analysis, 20 µL of essential oil (EO) was dissolved in 1480 µL of hexane. From the resulting solutions, 1 µL of each sample was injected into the GC system at an injection temperature of 250 °C.

Compound separation was performed on an AT-5MS capillary column (30 m length, 0.32 mm internal diameter, 0.25 µm film thickness). The chromatographic conditions were as follows: an initial temperature of 40 °C maintained for 2 min, followed by a temperature ramp to 250 °C at a rate of 4 °C/min, then a further increase to 300 °C at a rate of 10 °C/min, with a final hold at 300 °C for 5 min. The mobile phase was helium 6.0, with a flow rate of 1.92 mL/min. The interface temperature was set at 250 °C, and the ion source temperature at 210 °C.

Compound identification was carried out using mass spectrometry (MS) in scan acquisition mode (35–500 *m*/*z*), with spectral matching against the NIST database. Quantification was performed using the normalized area method. All samples were analyzed in triplicate.

Quantitative analyses were performed against retention index reference (RIr). For HPLC, identification of phenolic compounds was achieved by comparing retention times and UV-Vis spectra. The method was validated for semi-quantitative interpretation.

### 4.5. Sample Preparation for HPLC

One mL of GSE was diluted in 10 mL of a methanol–water solution (70:30, *v*/*v*) containing 0.1% formic acid. The mixture was subjected to sonication for 15 min at room temperature to enhance compound extraction. After sonication, the sample was centrifuged at 10,000 rpm for 10 min. The resulting supernatant was filtered through a 0.45 micrometer PTFE syringe filter, and the filtrate was subsequently injected directly into the HPLC system for analysis.

### 4.6. Instrumentation and Chromatographic Conditions

The analysis was performed using a High-Performance Liquid Chromatography (HPLC) system equipped with a UV-Vis photodiode array (PDA) detector. Separation of the compounds was achieved on a reverse-phase C18 column (e.g., 250 mm × 4.6 mm i.d., 5 µm particle size), such as the Agilent ZORBAX Eclipse Plus C18. The mobile phase consisted of a gradient elution using two solvents: Solvent A—water containing 0.1% formic acid, and Solvent B—acetonitrile containing 0.1% formic acid.

Peak assignment was performed by retention time and UV-spectral comparison. Data were collected and analyzed by Waters Millenium 32 software 3.03.05 (Milford, MA, USA) and tabulated by Microsoft Excel.

## 5. Conclusions

This study provides preliminary in vitro data supporting the potential antibacterial properties of GSE against multidrug-resistant bacteria isolated from the oral cavity of captive ball pythons. Of the substances tested, GSE exhibited the most consistent inhibitory activity, particularly on *Cryseobacterium* and *E. coli.* In contrast, the essential oils evaluated in this study did not demonstrate significant antibacterial effects under the conditions tested. These findings suggest that, within the limitations of in vitro testing, GSE may be a promising candidate for further investigation as a supportive or alternative antimicrobial agent in reptiles.

Nevertheless, the scope of the study remains exploratory. The absence of in vivo validation, pharmacokinetic data, and toxicologic profiles preclude any direct clinical recommendations. Future research should focus on evaluating the efficacy, safety, and appropriate formulations of GSE in reptilian models.

The characterization of antimicrobial resistance in exotic companion animals represents a critical endeavor within the One Health framework, given the close interconnection between animal health, human health, and the environment. These animals, often originating from diverse habitats and subjected to empirical or non-compliant therapeutic practices, may serve as unanticipated reservoirs of multidrug-resistant bacteria. Identifying and monitoring antimicrobial resistance profiles in these species not only optimizes veterinary therapeutic interventions but also helps prevent the transmission of resistant pathogens to humans through direct contact or shared environments. Integrating data from exotic companion animals into antimicrobial resistance surveillance enhances the relevance and accuracy of global strategies to combat this phenomenon. Therefore, the systematic investigation of these populations contributes to the development of evidence-based policies for the rational use of antimicrobials and reduces the risk of emergence of zoonotic pathogens.

## Figures and Tables

**Table 1 antibiotics-14-00549-t001:** List of bacterial species isolated and method of identification.

Identification	No of Isolates (n = 8)	Method of Identification	
		Initial	Confirmation
*Cryseobacterium idologenes*	3	API 20 NE	MALDI TOF-Ms
*Escherichia coli*	2	API 20E; Brilliance ESBL agar	MALDI TOF-Ms
*Staphylococcus epidermidis*	2	API Spaph	MALDI TOF-Ms
*Pseudomonas aeruginosa*	1	API 20 NE	MALDI TOF-Ms

**Table 2 antibiotics-14-00549-t002:** Results obtained with the disk-diffusimetric method.

Antimicrobials	Criteria for Categorization [23] S/I/R	Bacteria							
		1	2	3	4	5	6	7	8
		Diameter, mm (Categorization)							
Ampicillin	≥17/14–16/≤13	10 (R)	8 (R)	5 (R)	9 (R)	9 (R)	4 (R)	7 (R)	2 (R)
Cefaclor	≥20/–/≤16	5 (R)	0 (R)	7 (R)	11 (R)	9 (R)	13 (R)	14 (R)	8 (R)
Ceftiofur	≥21/18–20/≤17	12 (R)	10 (R)	6 (R)	14 (R)	11 (R)	14 (R)	15 (R)	11 (R)
Ciprofloxacin	≥25/19–14/≤18	15 (R)	13 (R)	9 (R)	17 (I)	18 (R)	13 (R)	8 (R)	8 (R)
Enrofloxacin	≥21/17–20/≤16	14 (R)	12 (R)	15 (R)	19 (I)	9 (R)	14 (R)	14 (R)	7 (R)
Penicillin	≥28/20–27/≤19	11 (R)	14 (R)	9 (R)	26 (I)	17 (R)	31 (S)	28 (S)	12 (R)
Amikacin	≥17/15–16/≤14	15 (I)	15 (I)	18 (S)	22 (S)	21 (S)	24 (S)	19 (S)	10 (R)
Doxycycline	≥14/11–13/≤10	18 (S)	15 (S)	20 (S)	22 (S)	13 (I)	25 (S)	22 (S)	7 (R)
Florfenicol	≥17/14–16/≤13	23 (S)	22 (S)	22 (S)	14 (I)	26 (S)	20 (S)	21 (S)	23 (S)
Sulfamethoxazole	≥16/11–15/≤10	11 (I)	18 (S)	23 (S)	19 (S)	26 (S)	22 (S)	18 (S)	19 (S)

**Legend:** S—sensitive; I—intermediate; R—resistant; 1–3—*C. indologenes*; 4–5—*E. coli*; 6–7—*S. epidermidis*; 8—*P. aeruginosa.*

**Table 3 antibiotics-14-00549-t003:** Phenotypic drug resistance profiles of the strain isolates from ball pythons.

Bacteria	Antibiotic Phenotype Patterns									
*C. indologenes*	AMP	CCL	EFT	CIP	ENR	P	AMK	DO	FFC	SXT
*C. indologenes*	AMP	CCL	EFT	CIP	ENR	P	AMK	DO	FFC	SXT
*C. indologenes*	AMP	CCL	EFT	CIP	ENR	P	AMK	DO	FFC	SXT
*E. coli*	AMP	CCL	EFT	CIP	ENR	P	AMK	DO	FFC	SXT
*E. coli*	AMP	CCL	EFT	CIP	ENR	P	AMK	DO	FFC	SXT
*S. epidermidis*	AMP	CCL	EFT	CIP	ENR	P	AMK	DO	FFC	SXT
*S. epidermidis*	AMP	CCL	EFT	CIP	ENR	P	AMK	DO	FFC	SXT
*P. aeruginosa*	AMP	CCL	EFT	CIP	ENR	P	AMK	DO	FFC	SXT

**Legend**: color red = resistant; color green = sensitive; color gray = intermediate; AMP—ampicillin, CCL—cefaclor, EFT—ceftiofur, CIP—ciprofloxacin, ENR—enrofloxacin, P—penicillin, AMK—amikacin, DO—doxycyclin, FFC—florfenicol, SXT—sulfamethoxazole.

**Table 4 antibiotics-14-00549-t004:** Antibacterial activity of essential oils/vegetal extract against bacterial strains isolated from pet ball pythons.

Essential Oils/ Vegetal Extract		*C. indologenes*	*E. coli*	*S. epidermidis*	*Ps. aeruginosa*
	Dilution	Inhibition Zone Diameter (mm)			
GSE	1/2	24	22	19	15
	1/4	23	21	17	13
	1/8	22	20	16	11
	1/16	21	19	15	10
	1/32	18	16	14	7
	1/64	15	14	13	NI
	1/128	8	7	12	NI
	1/256	7	5	9	NI
OEO	1/2	12	10	NI	NI
	1/4	8	NI	NI	NI
	1/8	NI	NI	NI	NI
	1/16	NI	NI	NI	NI
	1/32	NI	NI	NI	NI
	1/64	NI	NI	NI	NI
	1/128	NI	NI	NI	NI
	1/256	NI	NI	NI	NI
LGEO	1/2	NI	NI	NI	NI
	1/4	NI	NI	NI	NI
	1/8	NI	NI	NI	NI
	1/16	NI	NI	NI	NI
	1/32	NI	NI	NI	NI
	1/64	NI	NI	NI	NI
	1/128	NI	NI	NI	NI
	1/256	NI	NI	NI	NI
REO	1/2	NI	NI	NI	NI
	1/4	NI	NI	NI	NI
	1/8	NI	NI	NI	NI
	1/16	NI	NI	NI	NI
	1/32	NI	NI	NI	NI
	1/64	NI	NI	NI	NI
	1/128	NI	NI	NI	NI
	1/256	NI	NI	NI	NI
SEO	1/2	NI	NI	NI	NI
	1/4	NI	NI	NI	NI
	1/8	NI	NI	NI	NI
	1/16	NI	NI	NI	NI
	1/32	NI	NI	NI	NI
	1/64	NI	NI	NI	NI
	1/128	NI	NI	NI	NI
	1/256	NI	NI	NI	NI

**Legend**: color red = weak activity; color green = strong activity; color gray = moderate activity; NI = non-inhibition. GSE—grapefruit seed extract; OEO—Oregano essential oil; LGEO—Lemongrass essential oil; REO—Rosemary essential oil; SEO—Sage essential oil.

**Table 5 antibiotics-14-00549-t005:** Percentual composition of the essential oils (lemongrass, rosemary, sage, and oregano) determined by GS-MS.

Compound	Ric	RIr	Lemongrass %	Rosemary %	Sage %	Oregano %
Tricyclene	925	923	0.18	0.13		
*alpha*-Thujene	929	928		0.18		0.72
*alpha*-Pinene	934	936	0.26	11.95	0.28	2.06
Camphene	945	950	1.58	3.75	0.05	0.51
*beta*-Pinene	978	977		4.90	0.25	0.48
1-Octen-3-ol	979	980		0.07		0.13
5-Hepten-2-one, 6-methyl-	981	982	3.18			
*beta*-Myrcene	991	989		1.02	0.19	1.37
*alpha*-Phellandrene	1002	1004		0.15		
4-Carene	1010	1011		0.44		0.76
*p*-Cimene	1024	1025		1.10	0.04	14.94
Limonene	1030	1029	0.36	49.31	0.18	0.09
Eucalyptol	1031	1032				0.09
*gamma*-Terpinen	1058	1060		0.56		
4-Nonanone	1090	1093	1.72			5.85
Linalool	1098	1099	1.68	0.95	17.06	2.71
Camphor	1140	1143		14.61		
Carvomenthenal	1145	1148	0.25			
Citronellal	1152	1154	0.16			
Isoborneol	1157	1158	0.22			
Borneol	1165	1166		2.32		
*p*-Menth-2-ene	1168	1168	0.38			
p-Mentha-6,8-dien-2-ol,cis	1169	1170	0.19	0.58		
3-Cyclohexene-1-carboxaldehyde, 1,3,4-trimethyl-	1170	1171	0.46			
*alpha*-Terpineol	1188	1190	0.28	2.12	0.42	
Isopulegol acetate	1223	1225			0.12	
*beta*-Citral	1230	1232	28.26			
Thymol methyl ether	1233	1234				0.26
Linalyl anthranilate	1253	1255			77.34	
Geraniol	1255	1256	3.27			
*alpha*-Citral	1269	1270	43.43			
1,4-Hexadiene, 3-ethyl-4,5-dimethyl-	1273	1274			0.48	
Epoxy-linalooloxide	1275	1276	0.22			
Geranyl acetate, 2,3-epoxy-	1280	1281	0.29			
Bornyl acetate	1283	1283		0.58		
Thymol	1288	1290				3.60
Menthylacetate	1295	1297	1.05			
Carvacrol	1298	1300.4				64.70
Linalyl propanoate	1335	1336	0.27			
6-Hepten-3-ol	1360	1362	0.53			
Copaene	1368	1370			0.11	
Nerol acetate	1375	1376	6.91		0.82	
*beta*-Elemene	1389	1390.4	0.21			
Caryophyllene	1403	1406	0.77	4.60	1.02	0.95
*alpha*-Caryophyllene	1418	1420		0.44		
*gamma*-Cadinene	1512	1513	1.62			
*delta*-Cadinene	1515	1514	0.53			
Caryophyllene oxide	1578	1580	1.50	0.10	0.40	0.61
HM			2.75	73.78	1.49	20.93
OH			90.13	21.15	94.82	71.35
HS			3.13	4.60	0.93	0.95
OS			1.50	0.10	0.40	0.61
O			2.25	0.07		5.97

Legend: HM—hydrogenated monoterpenes; OH—oxygenated monoterpenes; HS—hydrogenated sesquiterpenes; OS—oxygenated sesquiterpenes; O—others (alcohols, aldehydes, non-terpene esters); RIc calculated retention indexes; RIr retention index references.

**Table 6 antibiotics-14-00549-t006:** Percentual composition of the grapefruit seed extract (GSE) determined by GS-MS.

Compound	RIc	RIr	%
Nonane	902	900	8.94243
Limonene	1030	1029	6.82735
Benzenemethanamine, N,N-dimethyl-	1352	1350	10.05916
3-Tetradecene, (Z)-	1384	1385	4.11613
1-Hexadecene	1492	1490	12.24403
1-Dodecanamine, N,N-dimethyl-	1529	1530	15.6739
1-Octadecene	1786	1790	5.06621
3(N,N-Dimethylmyristylammonio)propanesulfonate	1810	-	27.98891
1-Tridecanamine, N,N-dimethyl-	1865	1861	9.0819

Legend: RIc calculated retention indexes; RIr retention index references.

**Table 7 antibiotics-14-00549-t007:** Percentual composition of the grapefruit seed extract (GSE) measured by HPLC.

Name	Ret. Time	Concentration (μg/mL)
Caffeic acid	6.75	0.61
Ferrulic acid	7.18	0.50
Quercitin	16.15	1.15
Ascorbic acid	3.54	2.31
Cumaric acid	12.66	6.87
Naringin	12.5	3.82

## Data Availability

The data supporting this study’s findings are available from the corresponding authors upon reasonable request.

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
