# Peer review of "Antibacterial Activity of Some Essential Oils/Herbal Extracts Against Bacteria Isolated from Ball Pythons (Python regius) with Respiratory Infections"

_antibiotics, 2025, doi:10.3390/antibiotics14060549_

Round 1

Reviewer 1 Report

Comments and Suggestions for Authors

The manuscript addresses a relevant and relatively unexplored topic: the characterization of respiratory bacteria in pet pythons, including the evaluation of antimicrobial resistance and the potential use of plant extracts and essential oils as alternative therapies. The research question is valid, and the study fills a gap in the context of exotic reptile health.

However, the manuscript presents limitations in scope, structural organization, and scientific writing, which compromise its clarity, reproducibility, and overall impact. A section-by-section evaluation is provided below: 

ABSTRACT

Several phrases within the abstract present semantic redundancy or unnecessary lexical repetition, which detracts from the scientific clarity expected at this editorial level:

Lines 15–16:
The expressions “captive reptiles” and “these species” are used in close proximity with overlapping meaning. 

Suggestion:  “Respiratory diseases are among the main causes of morbidity and mortality in captive reptiles.”

Lines 17–18:
The phrase "key factors contributing to this issue" lacks precision; “this issue” should be replaced with a more specific reference, e.g., high incidence of pneumonia, to avoid ambiguity.

Line 19:
The clause bacteria... can directly cause the infection or worsen its progression is tautological. Suggest rephrasing to a more succinct statement, e.g., bacteria may act as primary pathogens or as facilitators of disease severity.”

Lines 29–31:
The list of antibiotics includes both ampicillin and penicillin, as well as two cephalosporins, all of which fall under β-lactam antibiotics. The sentence should be refined to avoid pharmacological overlap, for example: All isolates were resistant to β-lactams and fluoroquinolones.”

Line 33:
The term effective bioactivity is semantically redundant, as bioactivity inherently refers to biological effectiveness. Replace with: antibacterial activity.”

Line 38:
Phrases such as this study is timely and welcome introduce subjective judgment inappropriate for scientific abstracts. The abstract should maintain a neutral and evidence-based tone.

INDRODUCTION

Upon reviewing the Introduction section, I would like to present below a critical evaluation of its scientific merit, coherence, and alignment with the study objectives.

The Introduction outlines the general health concerns associated with respiratory infections in reptiles and the challenges posed by antimicrobial resistance. However, the scientific rationale lacks a clear focus, particularly in the initial paragraphs, where multiple themes—ranging from reptilian respiratory anatomy to zoonotic potential—are introduced without sufficient integration.

Lines 44–50:
These paragraphs mention infectious and parasitic diseases in reptiles, followed by anatomical and pharmacological considerations. While informative, the argument becomes fragmented. A more coherent synthesis is recommended, ideally transitioning from general to specific (i.e., from reptilian respiratory infections to the relevance of bacterial characterization in Python regius).

Lines 49–51:
The authors state that “in Romania, studies on herpetofauna health are in the early stages…”—this is a relevant point but would benefit from quantification or citation of national surveillance gaps to substantiate the claim.

The Introduction includes citations of key pathogens (Pseudomonas, E. coli, Chryseobacterium) and discusses their pathogenicity and zoonotic potential. However, the literature review approachs superficial in certain areas:

Lines 61–71:
The section discussing Pseudomonas aeruginosa introduces its dual role as commensal and pathogen, which is pertinent. Nonetheless, the sentence structure is lengthy and imprecise, and the citation of its virulence factors is mentioned without linking it to reptilian infections specifically.

Lines 72–90:
The inclusion of Chryseobacterium indologenes is appropriate and well-timed, given its rarity in reptile infections. However, the reference to a single case in Iran (line 75–76) would benefit from a more comprehensive synthesis of the pathogen's emerging profile in veterinary medicine.

The approach of Escherichia coli (lines 78–90) introduces various human-associated pathotypes (IPEC, ExPEC, APEC, NMEC), but their direct relevance to reptilian strains is not critically analyzed. The paragraph becomes a review of E. coli pathotypes rather than an argument for their study in snakes.

Despite covering relevant microorganisms and public health risks, the Introduction fails to articulate a clear research question or hypothesis. The transition from the literature review to the aim of the study is abrupt and lacks a formal objective statement.

Suggestion: the authors should state explicitly:
1 -  Why these bacteria were targeted;
2 -  Why Python regius was chosen;
3 - The novelty of testing essential oils and GSE against isolates from reptiles.

The Introduction section requires linguistic refinement for improved clarity and conciseness. Terms such as “have often been detected in water, soil, food…” (line 72) and “this has been demonstrated by the fact that…” (line 169) reflect a colloquial tone incompatible with high-impact scientific writing. Redundancies and awkward phrasing (e.g., “snakes may respond differently to treatment than mammals…” – line 51) should be addressed through careful editing.

RESULTS

Identification of Bacterial Strains (Lines 92–99) - The authors adequately report the use of phenotypic and proteomic methods (API systems and MALDI-TOF MS) for bacterial identification.

  • However, the sample size is critically limited (12 samples from 6 animals), which restricts the generalizability of the findings. The authors do not acknowledge this limitation within the Results or Discussion.
  • The proportional representation of isolates is correctly stated but not contextualized (e.g., relative to known prevalence in similar reptilian studies or expected flora).

Suggestion:: Include a table summarizing isolate frequency and identification method confirmation to enhance clarity.

Antimicrobial Susceptibility Testing (Lines 100–124) - The study provides phenotypic resistance profiles to ten antimicrobials. However, several critical issues are noted:

  • Absence of numerical values or summary statistics (e.g., inhibition zone diameters, mean, SD, or range) prevents the reader from interpreting variability or confidence in the data.
  • Resistance data are only discussed qualitatively and visualized using a color-coded scheme (red/green/grey). This format is incompatible with standard peer-review, especially as the version available lacks color coding. The interpretation is compromised as a result.
  • The classification of all isolates as "multidrug-resistant" is valid by standard definitions, but it should be explicitly defined (e.g., resistance to ≥3 antibiotic classes), with proper citation.

Suggestion: present resistance results in a numerical table format, include MIC or inhibition zone means where available, and cite EUCAST/CLSI criteria used for categorization. Color should be supplementary, not essential to interpretation.

Minimum Inhibitory Concentration of Essential Oils (Lines 125–175) - Table 2 summarizes the antibacterial activity of the tested essential oils and GSE across dilution gradients. However:

  • The zone of inhibition measurements are reported without standard deviation, raising questions about intra-test variability and reproducibility.
  • No positive or negative controls are mentioned in the table or text (e.g., vehicle control, standard antibiotics), which are essential to validate the inhibition observed.
  • The terminology used to describe effect strength ("strong," "moderate," "weak") is not supported by standardized criteria or reference values.
  • The authors claim that GSE had the most pronounced activity; however, they do not specify which concentrations meet established MIC thresholds, nor do they discuss potential cytotoxicity, solubility, or pH confounders associated with GSE use.

Suggestion: report MIC values where applicable, clarify inhibition zone categorization thresholds, and include discussion on variability. Reassess the use of qualitative descriptors without quantitative backing.

Essential Oil Composition (Tables 3–5) - The HPLC data for GSE are concise but not integrated into the antimicrobial discussion, which would be expected in a translational microbiological study.

Suggestion: include interpretative commentary connecting compound profiles to biological activity. This would strengthen the translational relevance of the chemical data presented.

In the results seccion the absence of statistical analysis is a major limitation. No measures of dispersion (mean ± SD), inferential testing, or replicates are mentioned. This undermines the robustness of the conclusions.

DISCUSSION 

The Discussion attempts to integrate a large volume of literature (lines 202–341), but does so without clear subsections or thematic progression. As a result, the argumentation becomes fragmented, and the central findings of the study are diluted within broader literature reviews.

Several paragraphs reiterate background information already addressed in the Introduction (e.g., zoonotic potential of C. indologenes, antimicrobial resistance in reptiles), rather than analyzing how the present study adds to that body of knowledge.

Suggestion: reorganize the Discussion using thematic anchors such as:

1 - Relevance of microbial findings (with emphasis on C. indologenes);

2 - Patterns of antimicrobial resistance and their implications;

3 - Comparative interpretation of essential oil and GSE efficacy;

4 - Practical implications and future directions;

5 - Study limitations.

Some specific points:

  • The detection of C. indologenes is presented as a novelty (lines 217–219), yet no discussion is offered regarding potential reservoirs, host specificity, or factors contributing to its emergence in reptiles; 
  • The multidrug resistance observed is acknowledged (lines 268–269), but there is no reflection on whether resistance profiles mirror anthropogenic antibiotic use, environmental contamination, or natural resistance traits;
  • The studies by Yeon-Sook Jho et al. (line 240) and Schmidt et al. (line 243) are briefly compared to the authors’ findings, but no hypotheses are presented to explain discrepancies (e.g., lack of Salmonella in this study); 
  • The Discussion includes subjective and informal expressions such as: “this study is timely and relevant” (line 345); “To our surprise...” (line 305); “we do not recommend the use of these products” (line 333). These phrases should be replaced by neutral, evidence-based statements in accordance with scientific writing norms.

MATERIAL AND METHODS 

Ethical Considerations and Sample Origin (Lines 357–365) - The clinical signs recorded are appropriate, but no mention is made of veterinary diagnostic confirmation (e.g., radiography, cytology) to corroborate pneumonia. It is important clarify how respiratory infection was diagnosed and whether criteria for inclusion were standardized. Indicate whether prior antibiotic exposure was ruled out, as this directly influences resistance outcomes.

Microbiological Isolation and Identification (Lines 367–385) - The MALDI-TOF MS is mentioned, the library or database used for species-level identification is not specified. Please, provide additional details regarding sample integrity control, bacterial culture conditions, and MALDI-TOF parameters (e.g., score thresholds, database version).

Antimicrobial Susceptibility Testing (Lines 386–399) - Please, clearly specify which interpretative criteria were used (e.g., CLSI M100 33rd Ed., EUCAST v.13.0), and address how non-standard host context was handled. Include replicate counts and basic descriptive statistics (mean, SD).

MIC and Essential Oil Testing (Lines 400–449) - It is important provide full disclosure of solvents/surfactants used, include details on control conditions, and report the number of replicates per treatment. Confirm whether antimicrobial activity was consistent across independent experiments.

GC-MS and HPLC Analyses (Lines 450–486) - Please, specify whether quantitative analyses were performed against reference standards, and whether the method was validated (even if partially) for semi-quantitative interpretation.

CONCLUSION

The authors claim (lines 488–498) that this is "the first study from our country to highlight the antibacterial action of essential oils (vegetal extracts) in the alternative treatment of diseases encountered in pet reptiles." Reframe such claims with appropriate caution. Suggestion: This study provides preliminary in vitro evidence supporting the potential antibacterial properties of GSE against multidrug-resistant bacteria isolated from pet pythons.”

Phrases such as “GSE showed a fairly intense action on all the tested bacteria” and “we note that only this vegetal extract showed bactericidal activity” lack quantitative support and scientific precision. The expression “we do not recommend the use of these products” (line 333, earlier in the discussion, but echoed here) reflects a prescriptive tone inappropriate in the absence of rigorous clinical validation. Replace subjective and informal formulations with neutral scientific language. Suggestion: “Among the tested substances, GSE exhibited the most consistent in vitro antibacterial activity.”

Although the authors briefly suggest that GSE could be explored as a treatment alternative, the conclusion lacks concrete future directions—a standard and expected component in this section. Add a forward-looking statement. Suggestion: “Future studies should evaluate the in vivo efficacy, pharmacokinetics, and safety profile of GSE and other natural products in reptilian models.”

In this reviewer's view, the conclusion reiterates findings but does not synthesize them in terms of biological significance or translational relevance. The authors could use the opportunity to emphasize the importance of characterizing antimicrobial resistance in exotic pets from a One Health perspective, which would enhance the relevance of their findings for both veterinary and public health. Additionally, the authors could close with a broader contextual statement linking the findings to zoonotic surveillance and responsible antimicrobial stewardship in non-conventional pets.

Comments on the Quality of English Language

The manuscript would benefit from a thorough revision by a fluent English speaker with experience in scientific writing. While the overall meaning is understandable, the text contains numerous grammatical inconsistencies, awkward phrasing, and instances of informal or subjective language that reduce its scientific clarity and professionalism. Specific issues include:

  • Recurrent use of redundant or imprecise expressions (e.g., “effective bioactivity,” “this study is timely and welcome”);
  • Inconsistent verb tenses and lack of subject-verb agreement in several sentences;
  • Use of colloquial language inappropriate for a scientific manuscript (e.g., “to our surprise,” “we do not recommend…”);
  • Wordiness and poor sentence structure in key sections (notably the Introduction and Discussion), affecting readability.

In my opinion, a professional English-language editing service or a revision by a native English-speaking colleague is strongly recommended to ensure that the manuscript meets the linguistic standards of the journal.

Reviewer 2 Report

Comments and Suggestions for Authors

Dear Authors,

in your manuscript you describe the isolation of bacteria from clinically ill snakes kept in a private collection. Also, you describe the results of the antimicrobial susceptibility of isolated bacteria and the sensitivity of the same bacteria to different essential oils. The positive side of this research is the relevance of the topic - antimicrobial resistance of bacteria isolated from exotic animals and the possible use of alternative preparations in the treatment of bacterial infections. The negative side is the lack of detailed description in the chapter Material and Methods that would enable the repetition of such research, the lack of relevant data in the Introduction, and not complete presentation of obtained results and discussions.

General comments:

Title should be changed as you also did investigate the AMR.

Please uniform writing of the bacterial scientific names in the whole text (for the first time full, later abbreviation); please check writing of Ps. aeruginosa (or P. aeruginosa)? 

Please uniform essential oils/herbal extract/plant extract/vegetal extract in the whole text

Introduction- only information about bacterial pathogens isolates are present - no scientific data on AMR in reptiles (snakes) or use of essential oils in the treatment of reptiles (snakes)

Results - a lot of text presented here could be more suitable for Discussion or Material and Methods (as L 102-104, L 166-175); a table could be composed to present the isolated by the single animal and sample; a table could be composed to present the results of multidrug resistant bacteria according to the AMR pattern; statistical analysis missing

Discussion – some of the sentences are more convenient for Conclusion; no discussion on the bacterial isolation from clinically ill snakes or on clinical symptoms? you detected the composition of essential oils by GC-MS and HPLC, but it is not mentioned in the Discussion? discussion on use of broth microdilution and microdilution in plates missing; chapters 4.5. and 4.6. should be completely changed

Materials and Methods- data on the animals missing (age, gender); data on samples missing; data on agar used for AMR missing

Specific comments.

L 22- please check spelling

L 40 – antibioresistance?

L 72-74 references missing

L 81-85 sentence is not clear enough, revision needed; E. coli not in italic

L 97 this agar not mentioned in Materials and Methods

L 110 phenicoli?

L 118 behavior of antibiotic?

Table 1 – intermediate and in L 392 only S or R?

L 128 oral cavity not mentioned as sample in Materials and Methods

Table 2- abbreviations used in the table not mentioned in the Legend

L 152 useless?

L 160 developed normally?

L 160-162 something missing? End of sentence?

Table 3 and Table 4 – please mentioned the methods used to obtain these results

L 280-282 sentence is not clear enough, revision needed (as well as L 314-316, L 323-325?

L 336-338 statement not clear – biochemical analysis of essential oils?

L 358-361 snakes were brought to the Faculty of Veterinary Medicine or the samples were taken at the breeder’s facility?

L 373-374 data on the use of API strips missing (according to the manufacturer recommendation?)

L 388 antibiotic behavior?

L 392-393 this sentence is not needed here

L 394-397 manufacturer of the discs missing

L 468-483 it looks like a text copied from a lab manual?

Comments on the Quality of English Language

 moderate revision needed

Round 2

Reviewer 1 Report

Comments and Suggestions for Authors

Considering the revisions made and in accordance with the updated text, I believe the manuscript is ready for publication.

Author Response

Dear Reviewer,

We are grateful to you for your thorough evaluation and insightful remarks, which have significantly enhanced the value of our manuscript. Your contribution is deeply appreciated.

Thank you,

The Authors

Reviewer 2 Report

Comments and Suggestions for Authors

Dear Authors,

thank you for taking into consideration the comments given by the reviewer. 

Regarding Table 1- did you mean the number of isolates (n=8) or the number of samples (n=8) from which you isolated the mentioned bacteria?

L 126-127 please rephrase, not clear enough (the definition of MDR?)

L 192 please check the end of the sentence

L 463 please change "It was no possible"

L 471-470 please remove the bullets;

L 479-480 please check this sentence

L 589, 592, 594 capitalized first letter missing

Comments on the Quality of English Language

minor editing needed

Author Response

Dear Reviewer,

We would like to sincerely thank you for your valuable comments and constructive suggestions, which have greatly contributed to improving the quality of our manuscript.

Below are the answers to your observations:

Regarding Table 1- did you mean the number of isolates (n=8) or the number of samples (n=8) from which you isolated the mentioned bacteria?

R: In Table 1 was mentioned the number of isolates from 12 samples (see line 110)

L 126-127 please rephrase, not clear enough (the definition of MDR?)

R: The phrase was rewritten 

L 192 please check the end of the sentence

R: resolved

L 463 please change "It was no possible"

R: was changed with impossible line 440

L 471-470 please remove the bullets;

R: resolved

L 479-480 please check this sentence

R: The phrase was rewritten

L 589, 592, 594 capitalized first letter missing

R: resolved

The authors